# UNSUPERVISED DOMAIN ADAPTATION THROUGH SELF-SUPERVISION

## ABSTRACT

This paper addresses unsupervised domain adaptation, the setting where labeled training data is available on a source domain, but the goal is to have good performance on a target domain with only unlabeled data. Like much of previous work, we seek to align the learned representations of the source and target domains while preserving discriminability. The way we accomplish alignment is by learning to perform auxiliary self-supervised task(s) on both domains simultaneously. Each self-supervised task brings the two domains closer together along the direction relevant to that task. Training this jointly with the main task classifier on the source domain is shown to successfully generalize to the unlabeled target domain. The presented objective is straightforward to implement and easy to optimize. We achieve state-of-the-art results on four out of seven standard benchmarks, and competitive results on segmentation adaptation. We also demonstrate that our method composes well with another popular pixel-level adaptation method.

## 1 INTRODUCTION

Visual distribution shifts are fundamental to our constantly evolving world. We humans face them all the time, e.g. when we navigate a foreign city, read text in a new font, or recognize objects in an environment we have never encountered before. These real-world challenges to the human visual perception have direct parallels in computer vision. Formally, a distribution shift happens when a model is trained on data from one distribution (source), but the goal is to make good predictions on some other distribution (target) that shares the label space with the source. Often computational models struggle even for pairs of distributions that humans find intuitively similar.

Our paper studies the setting of unsupervised domain adaptation, with labeled data in the source domain, but only *unlabeled* data in the target domain. The general philosophy of the field is to induce alignment of the source and target domains through some transformation. In the context of deep learning, a convolutional neural network maps images to learned representations in some feature space, so inducing alignment is done by making the distribution shifts small between the source and target in this shared feature space (Csurka, 2017; Wang & Deng, 2018; Gopalan et al., 2011). If, in addition, such representations preserve discriminability on the source domain, then we can learn a good classifier on the source, which now generalizes to the target under the reasonable assumption that the representations of the two domains have the same ground truth.

Most existing approaches implement this philosophy of alignment by minimizing a measurement of distributional discrepancy in the feature space (details in section 2), often some form of maximum mean discrepancy (MMD) e.g. Long et al. (2017), or a learned discriminator of the source and target as an approximation to the total variation distance e.g. Ganin et al. (2016). Both measurements lead to the formulation of the training objective as a minimax optimization problem a.k.a. adversarial learning, which is known to be very difficult to solve. Unless carefully balanced, the push and pull in opposite directions can often cause wild fluctuations in the discrepancy loss and lead to sudden divergence (details in section 2). Therefore, we propose to avoid minimax optimization altogether through a very different approach.

Our main idea is to achieve alignment between the source and target domains by training a model on *the same task* in both domains simultaneously. Indeed, if we had labels in both domains, we could simply use our original classification task for this. However, since we lack labels in the target domain, we propose to use a *self-supervised* auxiliary task, which creates its own labels directly from

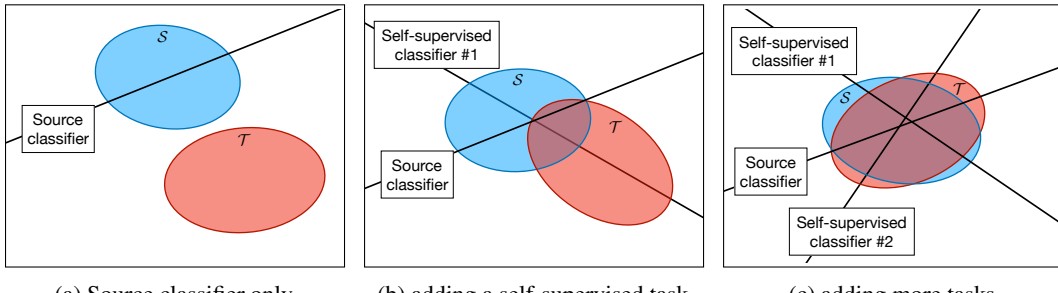

|  |  |  |
|---|---|---|
| (a) Source classifier only | (b) adding a self-supervised task | (c) adding more tasks |

Figure 1: We propose a method for unsupervised domain adaptation that uses self-supervision to align the learned representations of two domains in a shared feature space. Here we visualize how these representations might be aligned in the feature space. a) Without our method, the source domain is far away from the target domain, and a source classifier cannot generalize to the target. b) Training a shared representation to support one self-supervised task on both domains can align the source and target along one direction. c) Using multiple self-supervised tasks can further align the domains along multiple directions. Now the source and target are close in this shared feature space, and the source classifier can hope to generalize to the target.

the data (see section 3). In fact, we can use multiple self-supervised tasks, each one aligning the two domains along a direction of variation relevant to that task. Jointly training all the self-supervised tasks on both domains together with the original task on the source domain produces well-aligned representations as shown in Figure 1.

Like all of deep learning, we can only empirically verify that at least in our experiments, the model does not overfit by internally creating a different decision boundary for each domain along different dimensions, which would then yield bad results. Recent research suggests that stochastic gradient descent is indeed unlikely to find such overfitting solutions with a costly decision boundary of high complexity (implicit regularization), even though the models have enough capacity (Zhang et al., 2016a; Neyshabur et al., 2017b; Arora et al., 2018).

The key contribution of our work is to draw a connection between unsupervised domain adaptation and self-supervised learning. While we do not propose any fundamentally new self-supervised tasks, we offer insights in section 3 on how to select the right ones for adaptation, and propose in section 4 a novel training algorithm on those tasks, using batches of samples from both domains. Additionally, we demonstrate that domain alignment could be achieved with a simple and stable algorithm, without the need for adversarial learning. In section 5, we report state-of-the-art results on several standard benchmarks.

## 2 RELATED WORK

In this section we provide a brief overview of the two fields that our work would like to bridge.

### 2.1 UNSUPERVISED DOMAIN ADAPTATION

Methods for unsupervised domain adaptation in computer vision can be divided into three broad classes. The dominant class, which our work belongs to, aims to induce alignment between the source and the target domains in some feature space. This has been done by optimizing for some measurement of distributional discrepancy. One popular measurement is the maximum mean discrepancy (MMD) – the distance between the mean of the two domains in some reproducing kernel Hilbert space, where the kernel is chosen to maximize the distance (Bousmalis et al., 2016; Long et al., 2015; 2017). Another way to obtain a measurement of discrepancy is to train an adversarial discriminator that distinguishes between the two domains (Ganin & Lempitsky, 2014; Ganin et al., 2016; Tzeng et al., 2017). However, both MMD and adversarial training are formulated as minimax optimization problems, which are widely known, both in theory and practice, to be very difficult (Fedus et al., 2017; Duchi et al., 2016; Liang, 2017; Jin et al., 2019). Since the optimization landscape is much more complex than in standard supervised learning, training often does not converge or converges to a bad local minimum (Goodfellow, 2016; Nagarajan & Kolter, 2017; Li et al., 2017),

and requires carefully balancing the two sets of parameters (for minimization and maximization) so one does not dominate the other (Salimans et al., 2016; Neyshabur et al., 2017a).

To make minimax optimization easier, researchers have proposed numerous modifications to the loss function, network design, and training procedure (Arjovsky et al., 2017; Gulrajani et al., 2017; Karras et al., 2017; Courty et al., 2017; Sun & Saenko, 2016; Shu et al., 2018; Sener et al., 2016). Over the years, these modifications have yielded practical improvements on many standard benchmarks, but have also made the state-of-the-art algorithms very complicated. Often practitioners are not sure which tricks are necessary for which applications, and implementing these tricks can be bug-prone and frustrating. To make matters worse, since there is no labeled target data available for a validation set, practitioners have no way to perform hyper-parameter tuning or early stopping.

The second class of methods directly transforms the source images to resemble the target images with generative models (Taigman et al., 2016; Hoffman et al., 2017; Bousmalis et al., 2017). While similar to the first class in the philosophy of alignment, these methods operate on image pixels directly instead of an intermediate representation space, and therefore can benefit from an additional round of adaptation in some representation space. In subsection 5.2 we demonstrate that composing our method with a popular pixel-level method yields stronger performance than either alone.

The third class of methods uses a model trained on the labeled source data to estimate labels on the target data, then trains on some of those estimated pseudo-labels (e.g. the most confident ones), therefore bootstrapping through the unlabeled target data. Sometimes called self-ensembling (French et al., 2017), this technique is borrowed from semi-supervised learning, where it is called co-training (Saito et al., 2017; Zou et al., 2018; Chen et al., 2018; 2011). In contrast, our method uses joint training (of the main and self-supervised tasks), different from co-training in every aspect except the name.

## 2.2 Self-supervised Feature Learning

The emerging field of self-supervised learning uses the machinery of supervised learning on problems where external supervision is not available. The idea is to use data itself as supervision for auxiliary (also called "pretext") tasks that learn deep feature representations which will hopefully be informative for downstream "real" tasks. Many such auxiliary tasks have been proposed in the literature, including colorization (predicting the chrominance channels of an image given its luminance) (Zhang et al., 2016b; Larsson et al., 2017; Zhang et al., 2017), image inpainting Pathak et al. (2016), spatial context prediction (Doersch et al., 2015), solving jigsaw puzzles (Noroozi & Favaro, 2016), image rotation prediction (Gidaris et al., 2018), predicting audio from video (Owens et al., 2016), contrastive predictive coding (Oord et al., 2018), etc. Researchers have also experimented with self-supervision on videos (Wang & Gupta, 2015; Wang et al., 2019), and combining multiple self-supervised tasks together (Doersch & Zisserman, 2017).

Typically, self-supervision is used as a pre-training step on unlabeled data (e.g. the ImageNet training set without labels) to initialize a deep learning model, followed by fine-tuning on a labeled training set (e.g. PASCAL VOC) and evaluating on the corresponding test set. Instead, in this paper, we train the self-supervised tasks *together* with the main supervised task, encouraging a consistent representation that both aligns the two domains and does well on the main task[1].

Recently, self-supervision has also been used for other problem settings, such as improving robustness (Hendrycks et al., 2019), domain generalization (Carlucci et al., 2019) and few-short learning (Su et al., 2019). The most relevant paper to us is Ghifary et al. (2016), which uses self-supervision for unsupervised domain adaptation, but not through alignment. Their algorithm trains a denoising autoencoder Vincent et al. (2008) only on the target data, together with the main classifier only on the labeled source data. They argue theoretically that this is better than training the autoencoder on both domains together. However, their theory is based on the critical assumption that the domains are already aligned, which is rarely true in practice. Consequently, their empirical results are much weaker than ours, as discussed in section 5. Please see Appendix A for more detailed comparisons with these works.

---

[1]Note that it is possible to apply the standard pre-training followed by fine-tuning regime to unsupervised domain adaptation, doing self-supervised pre-training on the target domain (or both the source and the target) and then fine-tuning on the source. However, this gives almost no benefit over no-adaptation baseline, and is far from being competitive.

## 3 DESIGNING SELF-SUPERVISED TASKS FOR ADAPTATION

The design of auxiliary self-supervised tasks is an exciting area of research in itself, with many successful examples listed in section 2. However, not all of them are suitable for unsupervised domain adaptation. In order to induce alignment between the source and target, the labels created by self-supervision should not require capturing information on the very factors where the domains are meaninglessly different, i.e. the factors of variation that we are trying to eliminate through adaptation.

A particularly unsuitable tasks are these that try to predict pixels of the original image, as image in-painting (Pathak et al., 2016), colorization (Zhang et al., 2016b; Larsson et al., 2017; Zhang et al., 2017) or denoising autoencoder(Vincent et al., 2008). The success of pixel-wise reconstruction depends strongly on brightness information, or other factors of variation in overall appearance (e.g. sunny vs. coudy) that are typically irrelevant to high-level visual concepts. Thus, instead of inducing alignment, learning a pixel reconstruction task would instead serve to further separate the domains. We have experimented with using the colorization task and the denoising autoencoder for our training algorithm, and found their performance little better than the source only baseline, sometimes even worse! [2]

| Task | Images and self-supervised labels | | | |
|------|------|------|------|------|
| Rotation | 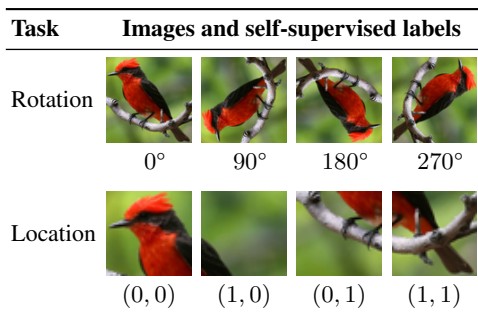 | | | |
| | 0° | 90° | 180° | 270° |
| Location | | | | |
| | $(0,0)$ | $(1,0)$ | $(0,1)$ | $(1,1)$ |

Table 1: Transformations and labels on a sample input image created by two of the self-supervised tasks used in our work.

In general, classification tasks that predict structural labels seems better suited for our purpose than reconstruction tasks that predict pixels. Therefore, we have settled on three classification-based self-supervised tasks that combine simplicity with high performance:

*Rotation Prediction:* (Gidaris et al., 2018) An input image is rotated in 90-degree increments i.e. 0°, 90°, 180°, and 270°; the task is to predict the angle of rotation as a four-way classification problem.

*Flip Prediction:* An input image is randomly flipped vertically; the task is to predict whether the image is flipped or not [3].

*Patch Location Prediction:* Patches are randomly cropped out of an input image; the task is to predict where the patches come from [4].

Consider a trivial illustrative example where the source and target are exactly the same except that the target pixels are all scaled down by a constant factor (e.g. daylight to dusk transition). All three of the aforementioned forms of self-supervision are suitable for this example, because pixel scaling i.e. brightness is "orthogonal" to the prediction of rotation, flip and location.

## 4 METHOD

Our training algorithm is simple once a set of $K$ self-supervised tasks are selected. We already have the loss function on the main prediction task, denoted $L_0$, that we do not have target labels for. Each self-supervised task corresponds to a loss function $\mathcal{L}_k$ for $k = 1...K$. So altogether, our optimization

---

[2]It is interesting to note that works borrowing from semi-supervised learning, e.g. Ghifary et al. (2016), use denoising autoencoder for their training algorithm and obtain performance better than source only (but still not competitive with ours). As explained in Appendix A, this difference is due to the difference in training algorithms – we use self-supervision on both domains while they only use on the target. This further reflects the difference in philosophy, as we use target data for alignment whereas they use it simply as extra data as in semi-supervised learning.

[3]We do not use horizontal flips, which is a common data augmentation technique, since it is typically desirable for natural scene features to be invariant to horizontal flips.

[4]For large images (e.g. segmentation), the crop comes from a continuous set of coordinates, and the task is a regression problem in two dimensions, trained with the square loss. For small images (e.g. object recognition), the crop comes from one of the four quadrants, and the task is a four-way classification problem; this distinction exists only for ease of implementation.

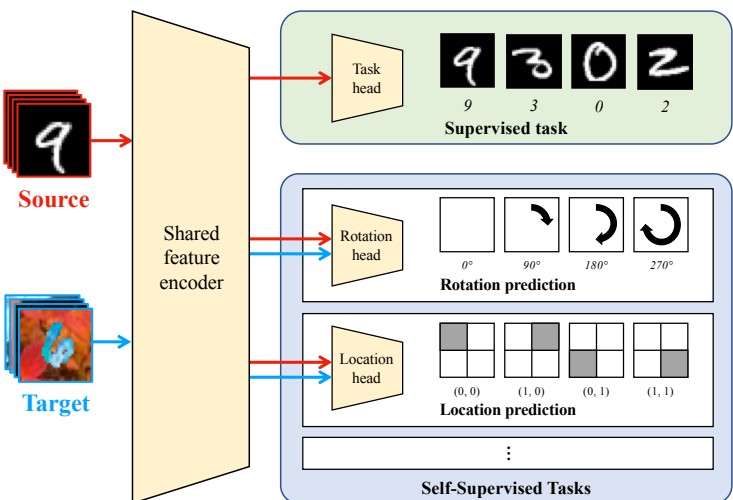

Figure 2: Our method jointly trains a supervised head on labeled source data and self-supervised heads on unbaled data from both domains. The heads use high-level features from a shared encoder, which learns to align the feature distributions.

problem is set up as a combination of these $K + 1$ loss functions, as is done for standard multi-task learning (see Figure 2). Implementation details are included in Appendix B.

Each loss function corresponds to a different "head" $h_k$ for $k = 0...K$, which produces predictions in the respective label space. All the task-specific heads (including $h_0$ for the actual prediction task) share a common feature extractor $\phi$. Altogether, the parameters of $\phi$ and $h_k, k = 0,..k$ are the learned variables i.e. the free parameters of our optimization problem.

In our paper, $\phi$ is a deep convolutional neural network, and every $h_k$ is simply a linear layer i.e. multiplication by a matrix of size output space dimension $\times$ feature space dimension. If $k$th task is classification in nature, then $h_k$ also has a softmax or sigmoid (for multi-class or binary) following the linear layer. The output space dimension is only four for rotation and location classification, and two for flip and location regression. Depending on the network architecture used, the feature space dimension ranges between 64 and 512. The point is to make every $h_k$ low capacity, so the heads are forced to share high-level features, as is desirable for inducing alignment. A linear map from the highest-level features to the output is the smallest possible head and performs well empirically.

Let $S = \{(x_i, y_i), i = 1...m\}$ contain the labeled source data, and $T = \{(x_i), i = 1...n\}$ contain the unlabeled target data. $L_0$ for the main prediction task takes in the labeled source data, and produces the following term in our objective:

$$\mathcal{L}_0(S; \phi, h_0) = \sum_{(x,y) \in S} L_0(h_0(\phi(x)), y).$$

Each self-supervised task $F_k$ for $k = 1...K$ modifies the input samples with some transformation $f_k$ and creates labels $\tilde{y}$. Denote $F_k(S) = \{(f_k(x_i), \tilde{y}_i), i = 1...m\}$ as the self-supervised samples generated from the source samples (with the original labels discarded), and $F_k(T) = \{(f_k(x_i), \tilde{y}_i), i = 1...n\}$ from the target. Then the loss $L_k$ of each task $k = 1...K$ produces the following term:

$$\mathcal{L}_k(S, T; \phi, h_k) = \sum_{(f_k(x), \tilde{y}) \in F(S)} L_k(h_k(\phi(f_k(x))), \tilde{y}) + \sum_{(f_k(x), \tilde{y}) \in F(T)} L_k(h_k(\phi(f_k(x))), \tilde{y}). \quad (1)$$

Note that $\mathcal{L}_k(S, T; \phi, h_k)$ for $k = 1...K$, unlike $\mathcal{L}_0(S; \phi, h_0)$, take in *both the source and target data*; as we emphasize for many times throughout the paper, this is critical for inducing alignment. Altogether, our optimization problem can be formalized as in multi-task learning [5]:

$$\min_{\phi, h_k, k=1...K} \mathcal{L}_0(S; \phi, h_0) + \sum_{k=1}^{K} \mathcal{L}_k(S, T; \phi, h_k). \quad (2)$$

---

[5]We have experimented with trade-off hyper-parameters $\lambda_i, i = 1...K$ for the loss terms inside the sum and found them unnecessary. Since a labeled target validation set might not be available, it can be beneficial to reduce the number of hyper-parameters.

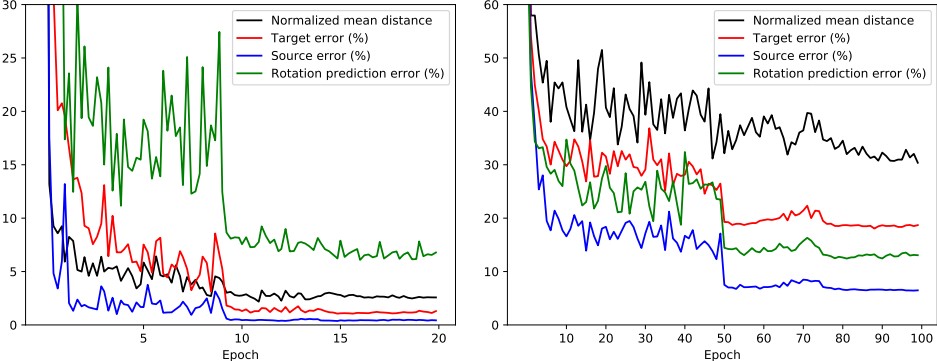

Figure 3: Results on MNIST→MNIST-M (left) and CIFAR-10→STL-10 (right). Test error converges smoothly on the source and target domains for the main task as well as the self-supervised task. This kind of smooth convergence is often seen in supervised learning, but rarely in adversarial learning. The centroid distance (linear MMD) between the feature distributions of the two domains converges alongside, even though it is never explicitly optimized for.

At test-time, we discard the self-supervised heads and use $h_0(\phi(x))$.

### 4.1 A HEURISTIC FOR HYPER-PARAMETER TUNING AND EARLY STOPPING

As previously mentioned, in the unsupervised domain adaptation setting, there is no target label available and therefore no target validation set, so typical strategies for hyper-parameter tuning and early stopping that require a validation set cannot be applied. This problem remains underappreciated; in fact, it is often unclear how previous works select their hyper-parameters or detemine when training is finished, both of which can be important factors that impact performance, especially for complex algorithms using adversarial learning. In this subsection we describe a simple heuristic [6], merely as rule-of-thumb to make things work instead of a technical innovation. Like almost all of deep learning, there is no statistical guarantee on this heuristic, but it is shown to be practically effective in our experiments (see Figure 3). We only hope that it serves as the first guess of a solution towards this underappreciated problem.

The main idea is that, because our method never explicitly optimizes for any measurement of distributional discrepancy, these measurements can instead be used for hyper-parameter tuning and early stopping. Since it would be counterproductive to introduce additional parameters in order to perform hyper-parameter tuning, we simply use the distance between the mean of the source and target samples in the learned representation space, as produced by $\phi$. Formally, this can be expressed as

$$D(S', T'; \phi) = \left\| \frac{1}{m} \sum_{x \in S'} \phi(x) - \frac{1}{n} \sum_{x \in T'} \phi(x) \right\|_2,$$   (3)

where $S'$ and $T'$ are *unlabeled* source and target validation sets [7].

Our heuristic combines $D(S', T'; \phi)$ and the main task error on the (labeled) source validation set. Denote $\mathbf{v} = (v_1, ..., v_T)$ and $\mathbf{w} = (w_1, ..., w_T)$ the measurement vectors of those two quantities respectively over $T$ epochs. The final measurement vector is $\mathbf{u} = \mathbf{v}/\min(\mathbf{v}) + \mathbf{w}/\min(\mathbf{w})$ i.e. a normalized sum of the two vectors; the epoch at which we perform early stopping is then simply $\arg\min_{t \in \{1...T\}} \mathbf{u}_t$. Intuitively, this heuristic roughly corresponds to our goal of inducing alignment while preserving discriminability.

---

[6] Appendix C contains some additional explanation of why this heuristic cannot be used for regular training. The short answer is given by the famous Goodhardt's law in economics and dynamical systems: "When a measurement becomes an objective, it ceases to be a good measurement."

[7] $D$ above is also known as the discrepancy under the linear kernel from the perspective of kernel MMD. Appendix C contains some additional explanation of why the mean distance is suitable for our heuristic, even though it is a specific form of MMD, which we claim to be difficult to optimize. The short answer is that it does not require minimax optimization.

| Source | MNIST | MNIST | SVHN | MNIST | USPS | CIFAR-10 | STL-10 |
| Target | MNIST-M | SVHN | MNIST | USPS | MNIST | STL-10 | CIFAR-10 |
|---|---|---|---|---|---|---|---|
| DANN (Ganin et al., 2016) | 81.5 | 35.7 | 73.6 | - | - | - | - |
| DRCN (Ghifary et al., 2016) | - | 40.1 | 82.0 | - | - | 66.4 | 58.7 |
| DSN (Bousmalis et al., 2016) | 83.2 | - | 82.7 | - | - | - | - |
| kNN-Ad (Sener et al., 2016) | 86.7 | 40.3 | 78.8 | - | - | - | - |
| PixelDA (Bousmalis et al., 2017) | 98.2 | - | - | - | - | - | - |
| ATT (Saito et al., 2017) | 94.2 | 52.8 | 86.2 | - | - | - | - |
| Π-Model (French et al., 2017) | - | 71.4 | 92.0 | - | - | 76.3 | 64.2 |
| ADDA (Tzeng et al., 2017) | - | - | 76.0 | 89.4 | 90.1 | - | - |
| CyCADA (Hoffman et al., 2017) | - | - | 90.4 | 95.6 | **96.5** | - | - |
| VADA (Shu et al., 2018) | 97.7 | 47.5 | 97.9 | - | - | 80.0 | 73.5 |
| DIRT-T (Shu et al., 2018) | **98.9** | 54.5 | **99.4** | - | - | - | 75.3 |
| VADA (IN) (Shu et al., 2018) | 95.7 | 73.3 | 94.5 | - | - | 78.3 | 71.4 |
| DIRT-T (IN) (Shu et al., 2018) | 98.7 | **76.5** | **99.4** | - | - | - | 73.3 |
| Source only VADA & DIRT-T | 58.5 | 27.9 | 77.0 | - | - | 76.3 | 63.6 |
| Source only VADA & DIRT-T (IN) | 59.9 | 40.9 | 82.4 | - | - | 77.0 | 62.6 |
| Source only our method | 44.9 | 30.5 | 92.2 | 94.7 | 81.4 | 75.6 | 58.8 |
| R | **98.9** | 61.3 | 85.8 [8] | 96.5 | 90.2 | **81.2** | 66.9 |
| R+L+F | - | - | - | - | - | **82.1** | **75.5** |

Table 2: Test accuracy (%) on standard domain adaptation benchmarks. Our results are organized according to the self-supervised task(s) used: R for rotation, L for location, and F for flip. We achieve state-of-the-art accuracy on four out of the seven benchmarks.

# 5 EXPERIMENTS

## 5.1 SEVEN BENCHMARKS FOR OBJECT RECOGNITION

The seven benchmarks are based on the six datasets described in Appendix D, each with a predefined training set / test set split, and labels are available on both splits. Previous works (cited in Table 2) have created those seven benchmarks by picking pairs of datasets with the same label space, treating one as the source and the other as the target, and training on the training set of the source with labels revealed and of the target with labels hidden. Following the standard setup of the field, labels on the target test set should only be used for evaluation, not for hyper-parameter tuning or early stopping; therefore we apply the heuristic described in subsection 4.1.

For the two natural scene benchmarks, we use all three tasks described in section 3: rotation, location and flip prediction. For the five benchmarks on digits we do not use location because it yields trivial solutions that do not encourage the learning of semantic concepts. Given the image of a digit cropped into the four quadrants, location prediction can be trivially solved by looking at the four corners where a white stroke determines the category. Adding flip to the digits does not hurt performance, but does not improve significantly either, so we do not report those results separately.

As shown in Table 2, despite the simplicity of our method, we achieve state-of-the-art accuracy on four out of the seven benchmarks. In addition, we show the source only results from our closest competitor (VADA and DIRT-T), and note that our source only results are in fact lower than theirs on those very benchmarks that we perform the best on; this indicates that our success is indeed due to effectiveness in adaptation instead of the base architecture.

Our method fails on the pair of benchmarks with SVHN, on which rotation yields trivial solutions. Because SVHN digits are cropped from house numbers with multiple digits, majority of the images have parts of the adjacent digits on the side. The main task head needs to look at the center, but the rotation head learns to look at the periphery and cheat. This failure case shows that the success of our method is tied to how well the self-supervised task fits the application. Practioners should use their domain knowledge evaluate how well the task fits, instead of blithely apply it to everything.

Also seen in Table 2 is that our method excels at object recognition in natural scenes, especially with all three tasks together. For STL-10→CIFAR-10, our base model is considerably worse than that of VADA and DIRT-T, but still beats all the baselines. Adding location and flip gives an improvement of 8%, on top of the 8% already over source only. This is not surprising since those tasks were

---

We will make our code public once the paper is accepted.

| | road | sidewalk | building | wall | fence | pole | traffic light | traffic sign | vegetation | terrain | sky | person | rider | car | truck | bus | train | motorbike | bicycle | mIoU |
|---|---|---|---|---|---|---|---|---|---|---|---|---|---|---|---|---|---|---|---|---|
| | | | | | | | | | GTA5 → Cityscapes | | | | | | | | | | | |
| Source only | 28.8 | 12.7 | 39.6 | 9.4 | 3.5 | 18.1 | 22.7 | 9.4 | 80.9 | 12.4 | 45.8 | 53.9 | 9.6 | 74.7 | 20.9 | 15.0 | 0.0 | 19.4 | 3.9 | 25.3 |
| Ours | 69.9 | 22.7 | 69.7 | 18.1 | 9.9 | 13.5 | 18.7 | 8.9 | 80.3 | 19.4 | 58.4 | 53.8 | 2.6 | 75.1 | 13.6 | 5.2 | 0.3 | 8.1 | 1.2 | 28.9 |
| CyCADA | 79.1 | 33.1 | 77.9 | 23.4 | 17.3 | 32.1 | 33.3 | 31.8 | 81.5 | 26.7 | 69.0 | 62.8 | 14.7 | 74.5 | 20.9 | 25.6 | 6.9 | 18.8 | 20.4 | 39.5 |
| Ours + CyCADA | 86.6 | 37.8 | 80.8 | 29.7 | 16.4 | 28.9 | 30.9 | 22.2 | 83.8 | 37.1 | 76.9 | 60.1 | 7.8 | 84.1 | 30.8 | 32.1 | 1.2 | 23.2 | 13.3 | 41.2 |
| Oracle | 97.3 | 79.8 | 88.6 | 32.5 | 48.2 | 56.3 | 63.6 | 73.3 | 89.0 | 58.9 | 93.0 | 78.2 | 55.2 | 92.2 | 45.0 | 67.3 | 39.6 | 49.9 | 73.6 | 67.4 |

Table 3: Test accuracy (%) on GTA5 → Cityscapes. Our method significantly improves over source only, and also over CyCADA when combined. This indicates that additional self-supervision using our training algorithm further aligns the domains.

originally developed for ImageNet pre-training i.e. object recognition in natural scenes – our method is very successful when the task fits the application well.

## 5.2 BENCHMARK FOR SEMANTIC SEGMENTATION

To experiment with our method in more diverse applications, we also evaluate on a challenging simulation-to-real benchmark for semantic segmentation – GTA5 → Cityscapes. GTA5 (Richter et al., 2016) contains 24,966 video frames taken from the computer game, where dense segmentation labels are automatically given by the game engine. Cityscapes (Cordts et al., 2016) contains 5,000 video frames taken from real-world dash-cams. The main task is to classify every pixel in an image as one of the 19 classes shared across both datasets, and accuracy is measured by intersection over union (IoU). The best possible results are given as the oracle, when the labels on Cityscapes are available for training, so typical supervised learning methods are applicable.

Our results are shown in Table 3, and implementation details in Appendix F. Our self-supervised tasks were designed for classification, where the label on an image depends on its global content, while in segmentation the labels tend to be highly local. Nevertheless, with very little modification, we see significant improvements over the source only baseline. In Appendix G we provide visualizations of the segmentation results, and make qualitative comparisons between those produced by the baseline and our method.

We also experiment with combining our method with another popular unsupervised domain adaptation method – CyCADA (Hoffman et al., 2017), designed specifically for segmentation. Surprisingly, when operating on top of images produced by this already very strong baseline, our method further improves performance. This demonstrates that pixel-level adaptation methods might still benefit from an additional round of adaptation by inducing alignment through self-supervision. We emphasize that these results are obtained with a very simple instantiation of our method, as a start towards the development of self-supervised tasks more suitable for semantic segmentation.

## 6 DISCUSSION

We hope that this work encourages future researchers in unsupervised domain adaptation to consider the study of self-supervision as an alternative to adversarial learning, and researchers in self-supervision to consider designing tasks and evaluating them in our problem setting. Most self-supervised tasks today were originally designed for pre-training and evaluated in terms of accuracy gains on a downstream recognition, localization or detection tasks. It will be interesting to see if new self-supervised tasks can arise from the motivation of adaptation, for which alignment is the key objective. Moreover, domain experts could perhaps incorporate their dataset specific knowledge into the design of a self-supervised task specifically for their application.

One additional advantage of our method, not considered in this paper, is that it might be particularly amenable to very small target sample size, when those other methods based on adversarial learning cannot accurately estimate the target distribution. We leave this topic for the future work.

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

APPENDIX

## A  ADDITIONAL DISCUSSION ON SPECIFIC PAPERS RELATED TO OURS

In this section we discuss the four papers mentioned in section 2 that are related to ours, but different in training algorithm, philosophy or problem setting.

*Deep Reconstruction-Classification Networks* (Ghifary et al., 2016). This method works in unsupervised domain adaptation, the same problem setting as ours. It learns a denoising autoencoder on the target, together with the main classifier on the source. However, they use only the target domain for reconstruction, claiming both empirically and theoretically that it is better than using both domains together. In contrast, we use both the source and target for self-supervision. These two algorithmic differences really reflect our fundamental difference in philosophy. They take the target data as analogous to the unlabeled (source) data in semi-supervised learning, so any form of self-supervision suitable for semi-supervised learning is good enough; their theoretical analysis is also directly borrowed from that of semi-supervised learning, which concludes that it is necessary and sufficient to only use the target data for self-supervision. We take data from both domains for self-supervision in order to align their feature distributions; also, both conceptually and empirically, we cannot use reconstruction tasks because they are unsuitable for inducing alignment (see section 3).

Empirical experiments further support our arguments in three ways: 1) If we were to only use the target for our self-supervision tasks, we would perform barely better than the source only baseline, even on MNIST→MNIST-M, the easiest benchmark where we would otherwise observe a huge improvement. 2) If we were to use the reconstruction task i.e. a denoising autoencoder, we would again perform barely better than the source only baseline, and sometimes even worse, as described in section 3. 3) As shown in Table 2, our results are much better than theirs on all of the benchmarks by a large margin. This shows that implementing the philosophy of alignment, developed for unsupervised domain adaptation, is much superior in our problem setting to blithely borrowing from semi-supervised learning.

In addition to the three classes of methods covered in section 2, unsupervised domain adaptation has also been studied using sample-to-sample correspondence through graph matching (Das & George Lee, 2018; Das & Lee, 2018); these methods are less relevant to ours and not discussed in detail.

*Using Self-Supervised Learning Can Improve Model Robustness and Uncertainty* (Hendrycks et al., 2019). This paper studies the setting of robustness, where there is no sample provided, even unlabeled, from the target domain. Their method jointly trains for the main task and the self-supervised task *only on the source domain*, really because there is no target data of any kind. Because their problem setting is very challenging (with so little information provided on the target), their primary evaluation is on a dataset of CIFAR-10 images with the addition of 15 types of common corruptions (e.g. impulse noise, JPEG compression, snow weather, motion blur and pixelation), simulated by an algorithm. No idea on unsupervised domain adaptation is mentioned.

*Domain Generalization by Solving Jigsaw Puzzles* (Carlucci et al., 2019). This paper studies two setting. The first is the robustness setting, exactly the same as in Hendrycks et al. (2019), except that evaluation is done on MNIST→MNIST-M and MNIST→SVHN. Their baseline is also a method from the robustness community that trains on adversarial examples and uses no target data. Again, because their problem setting is very challenging, the accuracy is low for both their proposed method and the baseline. The second setting is called domain generalization, which is very similar to meta-learning. The goal is to perform well simultaenously on multiple distributions, all labeled. Evaluation is done using the mean accuracy on all the domains. Beside the name, there is little similarity between the setting of unsupervised domain adaptation and domain generalization, which has no unsupervised component.

*Boosting Supervision with Self-Supervision for Few-shot Learning* (Su et al., 2019). As evident from the title, this paper studies the setting of few-shot learning, where the goal is to perform well on a very small dataset. Again, there is no unsupervised component and little connection to our setting.

Most of our results have already been produced when Carlucci et al. (2019) and Su et al. (2019) were presented at a conference. Hendrycks et al. (2019) was submitted to NeurIPS 2019 and should be considered concurrent work.

# B    ADDITIONAL ALGORITHMIC DETAILS

In practice and for our experiments, the source and target datasets are often imbalanced in size. If we were to blithely solve for the objective in Equation 2, the domain with a larger dataset would carry more weight for every $\mathcal{L}_k(S, T; \phi, h_k), k = 1...K$ because we are summing over all samples in both datasets. We would like to keep the two sums inside $\mathcal{L}_k$ roughly balanced such that in terms of features produced by $\phi$, neither of the two domains would dominate the other. This is easy to achieve in our implementation through balanced batches. When we need to sample a batch for task $k$, we simply sample half the batch from the source, and another half from the target, then put them together. In the end, our implementation requires little change on top of an existing supervised learning codebase. Each self-supervised task is defined as a module, and adding a new one only amounts to defining the structural modifications and the loss function.

We optimize the objective in Equation 2 with stochastic gradient descent (SGD). One can simply take a joint step on Equation 2, which covers all the losses for $k = 0$ and $k = 1...K$. However, the implementation would then have to store the gradients with respect to all $K + 1$ losses together. For memory efficiency, our implementation loops over $k = 1...K$; for each self-supervised task $k$, it samples a batch of combined source and target data (without their original labels), structurally modifies the images by $f_k$, creates new labels according to the modification, and obtains a loss for $h_k$ and $\phi$ on this batch. So a gradient step is taken for each $k = 1...K$, before finally a batch of original source images and labels are sampled for a gradient step on $h_0$ and $\phi$. In terms of test accuracy, these two implementations make little difference (usually less than 1%), and the choice simply comes down to a time-memory trade-off.

Because our source only baseline on STL-10 $\rightarrow$ CIFAR-10 is considerably worse than the closest competitor DIRT-T, we borrow from their implementation and use dropout regularization (Srivastava et al., 2014) with $p = 0.5$, only on this benchmark. This makes our source only baseline closer to theirs, so to allow for a fair comparison of the adaptation methods. Without this modification, our accuracy for source only, R, and R+L+F are respectively 56.1, 65.6 and 74.0%.

# C    ADDITIONAL DISCUSSION ON THE MEAN DISTANCE

In this section we answer some potential questions about our selection rule, which uses the mean distance, from the perspective of a possibly confused reader.

*Q*: The motivation of the paper is that methods based on minimax optimization, such as those using kernel-MMD, are difficult to optimize. Since you are using the mean distance (MMD under the linear kernel) for hyper-parameter tuning and early stopping, which is a form of model selection, how are you different from those other methods and why is optimization easy for you?

*A*: Our method is fundamentally different from those other methods based on MMD, and therefore more amenable to optimization in the following two ways: 1) Even though we use the mean distance, which is a form of MMD, we never pose a minimax optimization problem. The model parameters minimize the loss functions of the tasks, which we hope makes the mean distance small (see Figure 3). Model selection using subsection 4.1 also *minimizes* the mean distance. In our method, all the loss functions, for model parameters and hyper-parameters, work towards the same goal of inducing alignment, so optimization is easier. 2) Even though hyper-parameter tuning and early stopping are a forms of model selection just like training, there is a qualitative difference in the degrees of freedom involved. For subsection 4.1, when performing early stopping for example, optimization amounts to a grid search over the epochs after training is finished, and there is only one degree of freedom.

*Q*: Why is the mean distance suitable for model selection when it comes to hyper-parameter tuning and early stopping, but not regular training?

*A*: Again, we agree that hyper-parameter tuning and early stopping are a forms of model selection. However, unlike model parameters ranging in the hundred of thousands, there are at most a few hyper-parameters and only one parameter for early stopping. Model parameters can easily overfit to the mean distance while those few degrees of freedom we use cannot. This is precisely also the reason why previous works using MMD resort to minimax optimization.

# D  DETAILS OF THE SIX DATASETS USED FOR OBJECT RECOGNITION

1) MNIST (LeCun et al., 1998): greyscale images of handwritten digits 0 to 9; 60,000 samples in the training set and 10,000 in the test set.

2) MNIST-M (Ganin et al., 2016): constructed by blending MNIST digits with random color patches from the BSDS500 dataset Arbelaez et al. (2011); same training / test set size as MNIST.

3) SVHN (Netzer et al., 2011): colored images of cropped out house numbers from Google Street View; the task is to classify the digit at the center; 73,257 samples in the training set, 26,032 in the test set and 531,131 easier samples for additional training.

4) USPS: greyscale images of handwritten digits only slightly different from MNIST; 7291 samples in the training set and 2007 in the test set.

5) CIFAR-10 (Krizhevsky & Hinton, 2009): colored images of 10 classes of centered natural scene objects; 50,000 samples in the training set and 10,000 in the test set.

6) STL-10 (Coates et al., 2011): colored images of objects only slightly different from CIFAR-10; 5000 samples in the training set and 8000 in the test set.

Because CIFAR-10 and STL-10 differ in one class category, we follow common practice (Shu et al., 2018; French et al., 2017; Ghifary et al., 2016) and delete the offending categories, so each of these two datasets actually only has 9 classes.

# E  IMPLEMENTATION DETAILS ON THE OBJECT RECOGNITION BENCHMARKS

We use a 26-layer pre-activation ResNet (He et al., 2016) as our test-time model $h_0(\phi(x))$, where $h_0$ is the last linear layer that makes the predictions, and $\phi$ is everything before that. For unsupervised domain adaptation, there is no consensus on what base architecture to use among the previous works. Our choice is simply base on the widespread adoption of the ResNet architecture and the ease of implementation. In Table 2 we provide the source only results using our base architecture, and the ones from Shu et al. (2018), our closest competitor. Our source only results are in fact worse than theirs, indicating that our improvements are indeed made through adaptation.

At training time, the self-supervised heads $h_k, k = 1...K$ are simply linear layers connected to the end of $\phi$ as discussed in section 4. There is no other modification on the standard ResNet. For all experiments on the object recognition benchmarks, we optimize our model with SGD using weight decay 5e-4 and momentum 0.9, with a batch size of 128. We use an initial learning rate of 0.1 and a two milestone schedule, where the learning rate drops by a factor of 10 at each milestone. All these optimization hyper-parameters are taken directly from the standard literature (He et al., 2016; Huang et al., 2016; Guo et al., 2017) without any modification for our problem setting. We select the total number of epochs and the two milestones based on convergence of the source classifier $h_0$ and unsupervised classifiers $h_1, ..., h_K$. Early stopping is done using the selection heuristic discussed in subsection 4.1. For fair comparison with our baselines, we do not perform data augmentation, following previous works (Hoffman et al., 2017; Sener et al., 2016; Ghifary et al., 2016).

# F  IMPLEMENTATION DETAILS ON GTA5 → CITYSCAPES

For our experiments, we initialize our model from the DeepLab-v3 architecture (Chen et al., 2017), pre-trained on ImageNet, as commonly done in the field. Each self-supervised head consists of a global average pooling layer on the pre-logit layer, followed by a single linear layer. To take advantage of the large size of the natural scene images, we use the continuous i.e. regression version of location prediction. The self-supervised head is trained on the square loss, to regress the coordinates (in two dimensions) that the patch is cropped from. Natural for the regression version, instead of cropping from the quadrants like for the classification version on the small datasets, we instead crop out $400 \times 400$ patches taken at random from the segmentation scenes. We optimize our model with SGD using a learning rate of 0.007 for 15,000 iterations, with a batch size of 48. Once again, we use the selection heuristic in subsection 4.1 for early-stopping.

| Image | Ground truth label | Baseline prediction | Our prediction |
|---|---|---|---|

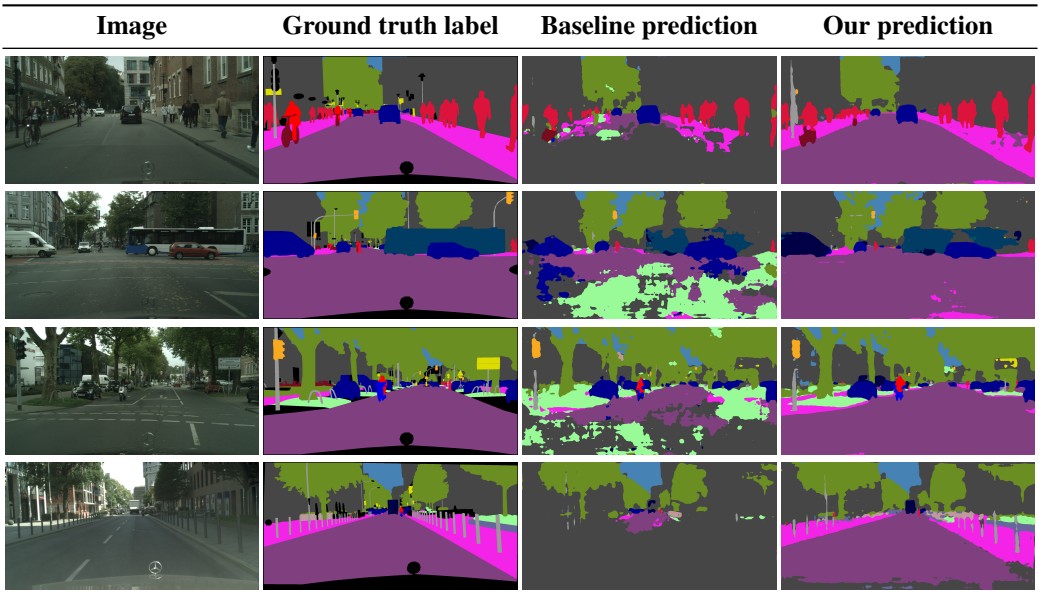

Table 4: Qualitative comparison of segmentation results using our method, the baseline, and the ground truth label.

## G    QUALITATIVE COMPARISON ON GTA5 → CITYSCAPES

We thank the anonymous Reviewer 3 for suggeting us to add these visualizations. This section shows segmentation results produced before and after adaptation(as shown in Table 4), alongside the original input image and the ground truth label.

Qualitatively, from visual inspection of these images, we can see that the quality of produced segmentations improves drastically after adaptation with our method. In particular, we note that location prediction as a self-supervised task helps to prevent nonsensical label configurations, which are often produced by the baseline. For example, the baseline model sometimes predicts "building" or "terrain" for pixels directly in front of a car. These errors are largely corrected for by our proposed adaptation method, leading to more accurate "road" predictions.

