# OpenReview forum: "Unsupervised Domain Adaptation through Self-Supervision"
_ICLR.cc/2020/Conference — Reject_

### Official Review · AnonReviewer2 · 2019-10-23
**Official Blind Review #2**

**Rating:** 6

**Review:**

This paper presents a novel unsupervised domain adaptation framework for neural networks. Similarly to existing approaches, it performs adaptation by aligning representations of the source and the target domains. The main difference is that this alignment is achieved not through explicitly minimizing some distribution discrepancy (this usually leads to challenging minimax optimization problems). Instead, the authors propose to use a battery of auxiliary self-supervised learning (SSL) tasks for both domains simultaneously. Each task is meant to align the source and the target representations along a direction of variation relevant to that task. Assuming that the battery is diverse enough, optimizing the representation for all the tasks leads to matching of the distributions.

Pros:
+ The paper is well-written and easy to read.
+ I like the simplicity of the idea and the fact that it achieves competitive performance without any adversarial learning (which may be very tricky to deal with).
+ The paper presents a reasonable procedure for hyper-parameter tuning and early stopping which seems to work well in practice.

Cons:
- The paper is purely practical with no theory backing the approach. As a result, the discussion of guarantees and limitations is quite brief.
- It’s unclear how easy it is to come up with a reasonable set of SSL tasks for a particular pair of domains. It seems that it may become a serious problem when the method is applied to something other than benchmarks. Table 2 reveals that there is no consistent improvement over the existing approaches which suggests that the chosen battery of SSL tasks is not universal (as the authors themselves admit). On a related note, it’s a bit disappointing that the authors mention SVHN results as a failure case but never provide a way to address the issue.
- It would be nice to some results for the Office dataset for completeness. The authors could use a pre-trained network as a starting points just like it’s done in other papers. According to the last paragraph of Section 6 this experiment should be feasible.

Notes/questions:
* Table 2, last column: The performance of DIRT-T seems to be better than that of the proposed method and yet the latter is highlighted and not the former.

Overall, I think it’s a good paper presenting a thought-provoking idea. In my opinion, the weakest point of the work is the lack of any (neither principled nor practical) guidance as to how to choose the set of self-supervised tasks. Despite this I feel that this submission should be accepted but at the same time I’m curious to see what the authors have to say regarding the concerns I raised in my review.

**Experience Assessment:**

I have published one or two papers in this area.

**Review Assessment: Checking Correctness Of Derivations And Theory:**

I assessed the sensibility of the derivations and theory.

**Review Assessment: Checking Correctness Of Experiments:**

I assessed the sensibility of the experiments.

**Review Assessment: Thoroughness In Paper Reading:**

I read the paper at least twice and used my best judgement in assessing the paper.

---

> ### Author Response · Authors · 2019-11-12
> **Thank you and answers to your questions**
>
> Thank you and we are happy you found our paper thought-provoking. Here we address the cons you wrote:
> 1. It is true that we have no theory backing our approach. On the other hand, it is rarely to see any deep learning paper with theory adequate enough to give “guarantees” for datasets we actually care about.
> 2. This connects well with the comment at the end of the review, asking us for “guidance as to how to choose the set of self-supervised tasks.” We have in fact given some practical guidance in the paper, which we summarize below as two necessary conditions:
> - The self-supervised task is well defined and nontrivial on both domains. This rules out the case of rotation prediction on SVHN, since as we explain in the paper, “the rotation head learns to look at the periphery and cheat”.
> - “The labels created by self-supervision should not require capturing information on the very factors where the domains are meaninglessly different.” as said and explained in section 3. This is rules out tasks such as colorization and autoencoder, for which it is important to learn the low-level details of the image.
> These two conditions are easy to reason about in practice. If the “battery of self-supervised tasks” satisfy them, there should be notable improvement on top of the source only baseline as we observe empirically, but there won’t be a guarantee. In addition, we would like this paper to add to the toolbox of available domain adaptation methods instead of becoming the only tool. When a good self-supervised task satisfying the two conditions cannot be found (SVHN), previous methods have provided different tools to use. When a good self-supervised task naturally exists, our method provides a simple and effective choice.
> In the end, this is a valuable question from the reviewer and we plan to be more explicit about those conditions in the next revision.
> 3. Please see results on Office-31 in our reply to R3.
>
> Your notes / questions: Thank you very much for pointing out our error with the highlighting. This is an honest typo. In the latest revision, we have improved our results to match that of DIRT-T; the modification we made for the improved results, as well as the original results, can be found in the last paragraph of Appendix B.

---

### Official Review · AnonReviewer1 · 2019-10-28
**Official Blind Review #1**

**Rating:** 3

**Review:**

This paper introduces an unsupervised domain adaptation method that uses self-supervised tasks to bring the two different domains closer together. It runs experiments on some classic benchmarks.

My score for this paper is weakly rejected because

(1) the concept of self-supervision is not first proposed by this paper. The proposed method is not novel. It introduces three simple self-supervision tasks: flip, rotation and location, and the performance is not better than previous results such as DIRT-T;

(2) there are 7 benchmarks in Table2, but only 2 of 7 has result on R+L+F. In the paper, it mentioned because the result is not better, but the author should still provide them.

(3) it emphasizes the contribution of encouraging more study of self-supervision for unsupervised domain adaptation. It doesn’t provide any way for how to design self-supervision task or whether more tasks is better. I think it is an interesting paper, but not enough as a conference paper, maybe a workshop paper.

(4) there are some classic unsupervised domain adaption benchmarks like Office Dataset, and Bing-Caltech dataset, why not run the method on them?

(5) In ICCV 2019, there is a paper "S4L: Self-Supervised Semi-Supervised Learning". The proposed method is almost same. I think the difference is this paper changes the setting and considers the unsupervised data as target domain and supervised data as source domain.

**Experience Assessment:**

I have published one or two papers in this area.

**Review Assessment: Checking Correctness Of Derivations And Theory:**

I assessed the sensibility of the derivations and theory.

**Review Assessment: Checking Correctness Of Experiments:**

I assessed the sensibility of the experiments.

**Review Assessment: Thoroughness In Paper Reading:**

I read the paper at least twice and used my best judgement in assessing the paper.

---

> ### Author Response · Authors · 2019-11-12
> **Thank you and answers to your questions**
>
> Thank you for your time giving us feedback. Here we answer your numbered concerns.
>
> 1. “The concept of self-supervision is not first proposed by this paper.” Since being proposed in the 1990s, self-supervised learning has become a wide and vibrant field of inquiry,  with hundreds, if not thousands of papers published in respected venues. So, we are perplexed by the statement:  is this arguing that all these papers were published in error?
> “The proposed method is not novel.”  Such statements are unhelpful without references to prior work. We have stated in the introduction what we perceive to be the novelties of our method. Please provide references to previously published papers that render our novelties invalid.
> “Performance is not better than previous results such as DIRT-T.”  Our results are shown in Table 2, and many of them are better than DIRT-T. Our method is also simpler and derived from a different perspective.
>
> 2. Below are the requested results for R+L+F:
> ————-————-————-————-————-————-—-————-————-
> Source		MNIST		MNIST		SVHN		MNIST		MNIST
> Target		MNIST-M	SVHN		MNIST		USPS		USPS
> ————-————-————-————-————-————-—-————-————-
> Accuracy (%)	98.7		        63.2		        85.7		        95.8		        87.0
> ————-————-————-————-————-————-—-————-————-
> There is not much difference between these numbers and the ones for R only.
>
> 3. “[The authors] do not provide any way for how to design self-supervision task”. Please see Section 3 titled “designing self-supervised tasks for adaptation”.
>
> 4. Please see results on Office-31 in our reply to R3.
>
> 5.  First, please note that ICCV 2019 papers are considered concurrent work, not prior work, to ICLR 2020 (ICCV’19 happened in November, whereas deadline for ICLR was in September).  Second, S4L, which is designed for semi-supervised learning, differs from ours both algorithmically and conceptually. We have already discussed this in the related work section in the context of semi-supervised learning methods, but to make our point clearer, here are the results for our implementation of the algorithm described in their equation (1) and (2) on MNIST -> MNIST-M, where improving upon the source only (no adaptation) baseline should have been very easy:
> ————-————-————-———
> 			        | Accuracy (%)
> ————-————-————-———
> Source only		|  44.9
> S4L method		|  56.6
> Our method		|  98.9
> ————-————-————-———
> The S4L result is barely better than source only, and qualitatively different from ours i.e. the difference should not come from merely implementation details. The most important difference between their algorithm and ours is that they train the supervised task on labeled data, and self-supervised task on unlabeled data, while we train the self-supervised task(s) simultaneously on both domains (labeled and unlabeled). Conceptually, training the self-supervised task on both domains is critical for alignment, which is the main objective for adaptation. Because for semi-supervised learning, the labeled and unlabeled data come from the same domain, methods for semi-supervised learning e.g. S4L do not need to consider the alignment problem. These comments are not intended to criticize S4L, as it is solving a different problem. In fact, theoretical analysis for semi-supervised learning [Cohen, Cozman] [Ghifary et al] suggests that training the self-supervised task on both domains is not helpful for semi-supervised learning; it is interesting to see how this picture is different for domain adaptation.
>
> “I think it is an interesting paper, but not enough as a conference paper, maybe a workshop paper.”   We are happy you found the paper interesting.  We do ask you to please reconsider your recommendation in light of the arguments presented above.  Also, as similar works using self-supervision as a tool, e.g. S4L, were published at respectable conferences instead of workshops, it seems reasonable to argue that this work, too, deserves to be accepted to ICLR.
>
> References:
> Cohen, I., Cozman, F.G.: Risks of semi-supervised learning: how unlabeled data can degrade performance of generative classifiers. In: Semi-Supervised Learning. MIT Press (2006)
> Muhammad Ghifary, W Bastiaan Kleijn, Mengjie Zhang, David Balduzzi, and Wen Li. Deep reconstruction-classification networks for unsupervised domain adaptation. In European Conference on Computer Vision, pp. 597–613. Springer, 2016.

---

### Official Review · AnonReviewer3 · 2019-10-28
**Official Blind Review #3**

**Rating:** 6

**Review:**

This paper describes an approach to domain adaptation that uses
self-supervised losses to encourage source/target domain alignment for
unsupervised domain adaptation. The authors propose to use four
self-supervised tasks (variants of tasks used in the self-supervised
representation learning for object recognition literature) that are
used with a combined loss including unlabeled source and target
training samples. The authors also propose an alignment heuristic for
guiding early stopping. Experimental results on a standard battery of
domain adaptation problems are given, plus some intriguing baseline
results for semantic segmentation.

The paper is written very well and the technical development and
motivations for each decision are well discussed and argued.

1. The experimental evaluation is a bit limited as the object
   recognition datasets are a bit limited. Results on Office or
   Office-Home would be nice.

2. Using location classification for semantic segmentation seems
   intuitively to be encouraging the network to learn coarse spatial
   priors (which should be invariant across the two domains). Have you
   looked at how alignment is actually happening? More qualitative
   analysis in this direction would be useful to appreciate the
   proposed approach.

3. Related to the previous point, it would be interesting to see how
   semgmentations in the unsupervised domain gradually change and
   improve with increasing alignment.

In summary: the ideas are simple, intuitive, and well-explained -- I
think the results reported would be easy to reproduce with minimal
head scratching. The experiments are interesting and not overstated.


**Experience Assessment:**

I have read many papers in this area.

**Review Assessment: Checking Correctness Of Derivations And Theory:**

I assessed the sensibility of the derivations and theory.

**Review Assessment: Checking Correctness Of Experiments:**

I assessed the sensibility of the experiments.

**Review Assessment: Thoroughness In Paper Reading:**

I read the paper at least twice and used my best judgement in assessing the paper.

---

> ### Author Response · Authors · 2019-11-12
> **Thank you and answers to your questions**
>
> Thank you for your thoughtful review. We have added qualitative comparisons in Appendix G of our latest revision (page 16).

---

### Public Comment · ~S._Alireza_Golestaneh2 · 2019-09-28
**How about the results on Office-Home dataset**

Interesting work! It would be nice to show the results on a more challenging dataset such as Office-Home as well, the provided datasets are very easy

---

> ### Author Response · Authors · 2019-10-03
> **Office very small**
>
> Office has an average of only 44 images per class per domain. Many other recent works e.g. many of our baselines do not use it because it is considered very small by the standard of modern deep learning.

---

### Public Comment · ~Researchers_CV1 · 2019-10-17
**The results of domain adaptation on semantic segmentation are weak**


It seems that the performance (28.9 and 41.2 with off-line transformed images) on semantic segmentation is weak compared to existing works.

Many works have more competitive performance;

Conditional Generative Adversarial Network for Structured Domain Adaptation, CVPR2018 (mIoU=44.5 with vgg19)
Fully Convolutional Adaptation Networks for Semantic Segmentation, CVPR2018
ROAD: Reality Oriented Adaptation for Semantic Segmentation of Urban Scenes, CVPR2018
DCAN: Dual Channel-wise Alignment Networks for Unsupervised Scene Adaptation, ECCV2018

---

> ### Author Response · Authors · 2019-10-17
> **Our point is not to have state-of-the-art segmentation results**
>
> We agree with this comment that recent works since 20`18 have better segmentation results. We would also like to emphasize that:
> - We are only claiming to improve segmentation results when our method is added on top of a prior work. Note that a separate self-supervised head can also be added to the prior works listed in the comment.
> - Segmentation is not the main result of the paper and only comprises a minor portion of our empirical section, while the methods listed above are explicitly designed for segmentation. In fact, all of our baselines in Table 1 have been accepted to major conferences without any result on segmentation, except CyCADA (which we do compare with on segmentation).

---

> > ### Public Comment · ~Researchers_CV1 · 2019-11-04
> > **The results of domain adaptation on classification are also weak**
> >
> > From table.2,  [Shu et al. 2018] performed better in most settings, including Mnist-->MnistM, Mnist->SVHN, SVHN->Mnist and STL-10 --> Cifar-10; although all these experiments were conducted on very small datasets, weak performance makes the story less convincing (Some hyper-parameters maybe the cause of little improvement).

---

> > > ### Author Response · Authors · 2019-11-04
> > > **This comment is based on incorrect readings of table 2**
> > >
> > > Unlike what the comment said, we perform worse than [Shu et al. 2018] only on the two benchmarks using SVHN. The datasets are standard in domain adaptation and far from "very small". In addition, [Shu et al. 2018] has been a state-of-the-art method. Comparing with this strong method does not make ours "weak". The anonymous commenter also attributes our improvements to hyper-parameter tuning without any evidence. Our method is not based on [Shu et al 2018] or any other baseline in table 2; we cannot make improvements just by tuning hyper-parameters.

---

### Public Comment · ~Kolo_Toure1 · 2019-11-08
**Regarding novelty and experiments**

First off, thanks for the work. I'd like to point out a few things that make me a little skeptical about this paper.

As blind reviewer 1 stated in [5], the method proposed here is quite similar to the ICCV19 paper "S4L: Self-Supervised Semi-Supervised Learning". I can't help but think that it is also similar to the CVPR19 "Domain Generalization by Solving Jigsaw Puzzles" (JiGen), which is basically the same as this paper, but in the domain generalization domain (with a different self-supervised task). I see that the authors have acknowledged this paper in the appendix. In fact, they state that the performance is bad, but you have to understand that it's the domain generalization setting, which makes me wonder - would this method present stronger performance that JiGen if experiments were done on the DG setting?

Also, I think experiements are quite sub-par. I've done some experiments on the "standard domain adaptation" benchmarks such as SVHN-MNIST, STL-CIFAR, and found that you can get large improvements by a combination of very small factors, such as data augmentation, network architecture, optimization method, and even hyperparameter tuning. This is largely due to the fact that SVHN-MNIST and CIFAR->STL is quite a simple task. In one of the comments below, the authors replied that the Office dataset is too "small by the standard of modern deep learning", but I'm not sure if SVHN-MNIST, CIFAR-STL is really any better. Yes, these are quite "large" datasets in terms of image quantity, but is adaptation of B&W digit images really any better for the standards of modern deep learning? I think this paper would be more convincing if experiments were conducted on larger datasets, such as the VisDA-2017 classification dataset. Frankly, I think this is a great dataset that is a much stronger reflection of real-world domain adaptation (much more so than any benchmarks containing MNIST or CIFAR), and there are plenty of previous works that have tested on this dataset to which the authors can compare their work with.

---

> ### Author Response · Authors · 2019-11-12
> **The criticisms are based on misunderstandings and lack justification**
>
> For the ICCV 2019 paper, please see our reply to reviewer 1 for a thorough comparison of the differences, both algorithmic and conceptual.
>
> For the domain generalization paper [Carlucci et al], we believe that you have misunderstood our words. We say in our paper that “because their problem setting is very challenging, the accuracy is low for both their proposed method and the baseline.” We are not criticizing Carlucci et al, but illustrating how their setting is different (as well as their method). You seem to think that we do not understand how their setting is different.
>
> We are then asked to perform experiments under the domain generalization setting of Carlucci et al. First, since this is a setting that our paper does not work on, these experiments are irrelevant to our purpose. Second, such experiments are in fact undefined. What does it mean to run a domain adaptation method in the domain generalization setting, where no target data is available in any form?
>
> You claim that our experiments are “quite sub-par”, because we could have obtained “large improvements by a combination of very small factors, such as data augmentation, network architecture, optimization method, and even hyperparameter tuning,” citing your own experience as evidence, without a publication, reference or code. First, there is no reason why our improvements are especially prone to such problems, in comparison to previous work on the benchmarks we use, such as DIRT-T published at ICLR 2018. We do not use data augmentation to keep a fair comparison with the baselines, the hyperparameters are set by our selection rule, and we use the default optimization method that comes with our network architecture, which is widely adopted in all of computer vision and without our method performs no better than the source only results of the baselines.
>
> Second, it is ambiguous what “large improvements” are - over source only (no adaptation) or the previous methods? If over source only, then the runs of our method share all the “small factors” mentioned in your comment as the runs of our source only, so the difference cannot be explained by these factors. If over the previous methods, this criticism is in fact undefined. Our method is not based on any of the baselines in Table 2, so does not even share their hyper-parameters; how can we then improve on them by hyper-parameter tuning?

---

### Decision · Program_Chairs · 2019-12-19

**Decision:**

Reject

**Comment:**

Thanks for your detailed replies to the reviewers, which helped us a lot to clarify several issues.
Although the paper discusses an interesting topic and contains potentially interesting idea, its novelty is limited.
Given the high competition of ICLR2020, this paper is still below the bar unfortunately.